# Future Prospects of Colorectal Cancer Screening: Characterizing Interval Cancers

**DOI:** 10.3390/cancers13061328

**Published:** 2021-03-16

**Authors:** Gemma Ibáñez-Sanz, Rebeca Sanz-Pamplona, Montse Garcia

**Affiliations:** 1Oncology Data Analytics Programme, Catalan Institute of Oncology, Hospitalet de Llobregat, 08907 Barcelona, Spain; gemma.ibanez@bellvitgehospital.cat; 2Gastroenterology Department, Bellvitge University Hospital, Hospitalet de Llobregat, 08907 Barcelona, Spain; 3Colorectal Cancer Research Group, ONCOBELL Programme, Institut d’Investigació Biomèdica de Bellvitge (IDIBELL), Hospitalet de Llobregat, 08907 Barcelona, Spain; 4CIBER Epidemiology and Public Health (CIBERESP), 28029 Madrid, Spain; 5Cancer Screening Unit, Prevention and Control Programme, Catalan Institute of Oncology, Hospitalet de Llobregat, 08907 Barcelona, Spain; 6Early Detection of Cancer Research Group, EPIBELL Programme, Institut d’Investigació Biomèdica de Bellvitge (IDIBELL), Hospitalet de Llobregat, 08907 Barcelona, Spain

**Keywords:** faecal immunochemical test, mass screening, screening programme, interval colorectal cancer, molecular characterization

## Abstract

**Simple Summary:**

The number of interval colorectal cancers is an intermediate measure of the effectiveness of a population-screening programme in decreasing colorectal cancer mortality. Differences between interval and screen-detected cancers have been reported. Better characterization of colorectal cancer to differentiate false-negative faecal immunochemical test results from true interval cancers would help determine whether to focus more on the organizational aspects of a screening programme or on stratifying colorectal cancer risk. This review provides a comprehensive overview of the epidemiological, clinical, and molecular characteristics of interval colorectal cancers.

**Abstract:**

Tumors that are not detected by screening tests are known as interval cancers and are diagnosed clinically after a negative result in the screening episode but before the next screening invitation. Clinical characteristics associated with interval colorectal cancers have been studied, but few molecular data are available that describe interval colorectal cancers. A better understanding of the clinical and biological characteristics associated with interval colorectal cancer may provide new insights into how to prevent this disease more effectively. This review aimed to summarize the current literature concerning interval colorectal cancer and its epidemiological, clinical, and molecular features.

## 1. Introduction

Colorectal cancer (CRC) screening based on faecal occult blood followed by colonoscopy has been demonstrated to reduce CRC mortality [1]. Unlike other screenings, a premalignant polyp can be detected and removed in some cases; thus, the incidence of CRC can be decreased, albeit to a limited extent [2]. Most organized screening programmes currently use the faecal immunochemical test (FIT) as the preferable screening method for CRC [2,3]. The goal of a screening programme is to detect the largest number of existing tumors in the population before their clinical occurrence. Consequently, a key performance indicator of an organized population-based screening programme is the monitoring of interval cancers. Although interval cancers are inevitable in a screening programme, their number should be as small as possible because a high proportion would decrease screening effectiveness. Identification of clinical and molecular characteristics associated with interval CRC, in contrast to screen-detected CRC, may provide new insights into how to prevent interval CRC more effectively through improved risk stratification.

### 1.1. Interval Colorectal Cancer Definition

A screen-detected cancer is a cancer detected by endoscopic confirmation after a positive test. By contrast, the World Endoscopy Organisation defines an interval CRC as that diagnosed clinically after a negative test result in a screening episode and before the next invitation to the programme [4] (Figure 1). In an FIT screening programme, interval CRC can appear after a negative FIT or after a positive FIT followed by a negative colonoscopy. An FIT interval CRC is a cancer detected after a negative faecal occult blood screening test and before the next invitation is due (2 years). Post-colonoscopy CRC (PCCRC) is a cancer diagnosed after a positive faecal occult blood test followed by a colonoscopy without CRC and before the next recommended screening or surveillance examination.

Recently, new guidelines on post-polypectomy surveillance have been published [5,6,7], and all share the aim towards more selective and less frequent surveillance. Currently, surveillance colonoscopy after three years is recommended in patients with “high-risk findings”, comprising at least 1 adenoma ≥ 10 mm or with high-grade dysplasia, ≥5 adenomas, or any serrated polyp ≥ 10 mm or with dysplasia (Figure 1). By contrast, patients with “no high-risk criteria” do not require endoscopic surveillance and should be returned to screening. These guidelines are based on recent evidence [8,9,10,11,12] showing that the long-term CRC risk in patients with non-advanced neoplasia is not higher than that in the general population. If organized screening is not available, the repetition of colonoscopy 10 years after a negative colonoscopy is recommended [13,14]. However, these guidelines have been recently published; no previous standard definition for post-colonoscopy CRC was available.

An interval CRC could arise because of the following [4]: a missed lesion (prior examination adequate or prior examination negative but inadequate); a detected lesion not resected or incompletely resected; failed biopsy detection; or a de novo rapid growth CRC).

In this review, we will focus on FIT interval cancer because of the greater uniformity in the definition [15]. Given the known natural history of CRC, most interval cancers detected within one or two years are likely due to a false-negative FIT result (Figure 1). However, if an advanced preneoplastic lesion is missed by the FIT, a much shorter sequence for CRC is possible. In this case, it will not be easy to decide whether to classify them as missed lesions or new cancers.

### 1.2. Diagnostic Accuracy of Faecal Immunochemical Tests in CRC Screening

In a population-screening context, the FIT has a sensitivity of 86% and a specificity of 85% to detect CRC [16] and a sensitivity of 29% and a specificity of 97% to detect advanced adenoma [17]. Thus, the FIT is much less sensitive for advanced adenomas. However, the natural history of these lesions to CRC and programmatic screening are opportunities to detect them [18]. The sensitivity is significantly higher for polyps with advanced histology, a larger size, and a pedunculated shape, and lower for serrated lesions [19,20,21,22,23,24,25,26,27,28]. Some studies [29,30,31] have evaluated the differences in the diagnostic performance of the FIT in detecting neoplasia located in the proximal or right-sided colon (colon above the level of the splenic flexure) versus the distal colon. The most recent meta-analysis [29] reported that the FIT had a lower sensitivity for detecting advanced adenomas located in the proximal colon than for detecting lesions in the distal colon compared with colonoscopy. By contrast, the sensitivity for detecting CRC located in the distal colon was comparable to that in the proximal colon, unlike a previous meta-analysis study [30]. In a primary study with a symptomatic population, no differences were found in the CRC sensitivity by location [32].

### 1.3. Episode Sensitivity in an FIT Screening Programme

CRC screening is intended to reduce CRC cancer mortality by detecting neoplasia at an earlier stage. High sensitivity is needed for an FIT screening programme to fulfil its purpose. Thus, the programme should have the minimum number of interval cancers. The sensitivity of an FIT screening programme can be determined by two methods: (1) interval cancer proportion (detection method), which calculates the proportion of interval cancers among all cancers; (2) proportional interval cancer rate (incidence method), which contrasts the incidence of interval cancers with the expected population incidence rate without screening. European Guidelines for Quality Assurance in CRC [33] recommend using the detection method. However, this method compares screen-detected cancers, which can become symptomatic in a longer period than two years, with interval cancers, which are diagnosed within two years by definition. Thus, the detection method can overestimate the episode sensitivity, particularly when measuring a prevalent screening round. By contrast, the incidence method, which is not affected by overdiagnosis or duration bias, can be difficult to calculate because it uses the incidence rate without screening.

The objective of this article was to review the evidence of the epidemiological, clinical and molecular characteristics of interval cancers in CRC. First, we reviewed the interval CRC proportion identified in FIT population-based screening programmes and summarized patient-related characteristics associated with interval cancer. Second, we summarized all the evidence concerning the molecular characterization of interval CRC.

## 2. Methods

### Search Strategy and Selection Criteria

For the first objective, reviewing the evidence of the epidemiological and clinical characteristics of interval CRCs in population-based screening programmes, we used a precision-maximizing search strategy using PubMed until December 2020 (https://pubmed.ncbi.nlm.nih.gov/; accessed on 28 December 2020) to retrieve studies of interval cancers after negative immunochemical testing in organized screening programmes. The search terms included were as follows: (FIT* OR fecal immunochemical OR faecal immunochemical OR immunochemical test) AND (mass screening OR population-based screening program* OR organized screening program* OR programmatic screening) AND (“interval cancer” OR “interval colorectal cancer”) AND (colorectal neoplasms OR colorectal cancer OR bowel cancer). Twenty-three articles were individually reviewed to retrieve studies that reported CRC occurrence within 1–2 years after a negative FIT in average-risk screening populations. Additionally, the reference lists of selected studies were revised to identify additional relevant studies. We excluded studies with guaiac faecal occult blood tests, with fewer than 100 screen-detected CRC samples or with insufficient data to calculate the CRC interval proportion. The study quality score of the 12 observational studies was analysed according to the Newcastle-Ottawa criteria [34] (Appendix A).

For the second objective, reviewing papers describing the molecular features of interval CRC, a literature search was conducted using PubMed until December 2020. The search terms included were as follows: (“interval colorectal cancer” OR postcolonoscopy OR post-colonoscopy OR “interval cancer” OR I-CRC) AND colorectal cancer AND screening AND (molecular OR genetic OR genomic OR biological OR mutation). Eighty abstracts were individually reviewed to retrieve studies that determined the biological characteristics of interval cancers (FIT or post-colonoscopy interval cancers). Articles were carefully evaluated, and reference lists were examined to include further appropriate publications. We excluded studies conducted only in hereditary cancer populations. The study quality score of the 13 observational studies was analysed according to the Newcastle–Ottawa criteria [34] (Appendix A).

## 3. Results

### 3.1. Interval CRC Proportion and Epidemiological and Clinical Characteristics of Interval CRC 

Wieten et al. [35] published a systematic review and meta-analysis to determine the incidence rates of interval CRC following the guaiac faecal occult blood test and FIT in population-based CRC screening programmes. They found a higher incidence rate of interval cancer after a negative guaiac faecal occult blood test than after the FIT (34 vs. 20 interval CRCs per 100,000 person-years, respectively). Moreover, they reported that for every FIT interval CRC, 2.6 screen-detected cancers were found. However, scarce information on interval cancer characteristics was provided.

After a systematic literature search, 12 studies which reported interval CRC after a negative FIT in an average-risk population were included, comprising 13,760,868 screening participants overall. The quality of most studies was judged as high (Appendix A), and all were cohort studies. The characteristics of the included studies are shown in Table 1. More than half of the studies used a 20-μg Hb/g faeces faecal haemoglobin positivity cut-off. The interval CRC proportion identified in screening using the FIT ranged from 4.6% to 59.1%, and the proportional interval cancer rate ranged from 4.0% to 25.0% (Table 1). FIT sensitivity might differ among studies because of, different cut-off values set for test positivity, screening participation, study population, FIT brand [36], and number and type (prevalent or incident) of screening rounds analysed. The interval cancer proportion was higher in the first round of the FIT and improved in subsequent rounds. The explanation for this finding is that, with every subsequent round, more cancers are diagnosed; consequently, fewer CRCs are missed with the FIT. The global efficacy of an FIT-based CRC screening programme depends on the cumulative sensitivity of repeated tests every two years [31].

Table 1 shows that nine of the twelve studies found that FIT interval CRC was more frequently located in the proximal colon, a finding that is consistent with previous studies [35,49]. FIT interval cancers are approximately twice more likely to occur in the proximal colon than in the distal colon compared with screen-detected CRCs [31]. Nevertheless, Buron et al. [44] reported an increased incidence of interval CRC located not only in the right colon but also in the rectum, likely because rectal bleeding is a symptom that alerts the patient. Several reasons explain the difference in FIT sensitivity for right-sided versus left-sided cancers. First, and as already mentioned above, these findings may be explained by the morphology and characteristics of the proximal lesions (sessile or flat lesions), which may be associated with less bleeding. Second, the FIT detects adenomas, which have a larger size and pedunculated shape (frequently located in the distal colon) than sessile serrated adenomas, which are usually flat and smaller (frequently located in the proximal colon). Finally, faecal haemoglobin originating from a proximal lesion could be subjected more to degradation during bowel passage, causing more false-negative results.

Six of ten studies that analysed differences in cancer incidence according to sex reported a higher risk of interval cancer in women. Previous studies have already reported lower test sensitivity in women [50,51]. Brenner et al. [50] observed that, at any cut-off of the quantitative FIT, the sensitivity and positive predictive value for detecting advanced neoplasia were substantially lower among women, partly explained by sex differences in the prevalence of advanced colorectal neoplasia. Van Turenhout et al. [51] evaluated whether sex differences in FIT sensitivity existed for CRC. They found that women had a lower FIT sensitivity and a higher specificity for CRC than men. The reason was likely due to the presence of larger and more advanced adenomas in men than in women (because it increases the probability of bleeding). In this study, sex differences were not explained by age or tumor location. By contrast, previous studies have suggested that right-sided CRCs occur more frequently in women [52,53,54]. Hormonal factors and genetic changes [55,56,57], such as sporadic microsatellite instability that occurs with ageing [58,59], may contribute to a higher prevalence of right-sided CRC. Finally, these sex differences in sensitivity could also be explained by the slower colonic transit time in women [60,61,62].

Interval CRC was also associated with advanced age in two of the studies. The reason could be explained by elderly individuals having a higher risk for colorectal neoplasia [63,64] and a higher frequency of right-sided neoplasia [65,66]. A discrepancy exists in the long-term trend of detection rates between proximal neoplasia [31], likely due to failure of the FIT to detect the increased presence of advanced adenomas in the proximal colon at older ages, or proximal advanced lesions progressing to invasive CRC faster in older patients.

Although female sex and advanced age are associated with right-sided CRC, a recent systematic review reported that the prevalence of sessile serrated polyps did not differ by sex or age [67].

Finally, interval CRC was staged as more advanced in six of the studies, likely because of more aggressive tumors. The advanced stage and higher mortality observed in interval cancers could be explained by the effect of a delayed diagnosis mostly caused by false-negative FIT results [39,68]. However, the worse prognosis associated with true interval cancers could be related to fast-growing neoplasia.

The haemoglobin concentration influences risk prediction of an interval cancer [69,70]. However, in four of the studies, false negatives had an almost undetectable concentration of faecal haemoglobin [42,43,44,45,46]. Similar to other previously reported studies [71], these false-negative lesions would be missed regardless of the cut-off value for positive FIT results. Thus, it is important to inform our target population to carefully monitor the symptoms and avoid patients being falsely reassured by the FIT result [72].

### 3.2. Molecular Characterization of Interval CRC

#### 3.2.1. Pathways of Tumorigenesis and Consensus Molecular Subtype (CMS) Classification

CRC is a heterogeneous disease comprising different molecular entities. Indeed, three features classify CRC tumors from a molecular perspective [73]: (1) chromosomal instability (CIN) or the conventional adenoma-carcinoma pathway, (2) microsatellite instability (MSI) and (3) the serrated polyp pathway exemplified by CpG island methylation ([CIMP]-high) and *BRAF* mutation.

Microsatellite stable (MSS) tumors harbour chromosomal abnormalities, whereas MSI tumors are hypermutant neoplasias. Both MSI and CIMP phenotypes can coexist in the same tumor [74]. More recently, the CRC Subtyping Consortium defined novel CRC molecular subtypes (CMSs) based on gene expression data (Figure 2). CMS1 (MSI immune) includes tumors with a better prognosis that are associated with MSI and have immune cell infiltration. CMS2 tumors (canonical) are characterized as microsatellite stable and have activated WNT/MYC pathways. CMS3 (metabolic) tumors are mainly characterized by activated pathways related to metabolism. Finally, CMS4 tumors (mesenchymal) show elevated TGF-β signalling and stromal infiltration.

#### 3.2.2. Studies of Molecular Characterization of Interval CRC

Although many interval cancers are considered lesions that were not detected at previous colonoscopies or a previous FIT, few studies have investigated interval tumor molecular characterization. To our best knowledge, only one study that has investigated the molecular features of interval CRCs using guaiac faecal occult blood tests and none using the FIT. We excluded studies performed only in patients with hereditary cancer. After a systematic literature search, we included 13 studies that evaluated the biological and genomic characteristics of interval CRCs (Table 2). All the studies were performed after a diagnostic colonoscopy except Nishihara et al. [75] where the colonoscopy was a screening test and Walsh et al. [49] performed in a guaiac-based screening programme. The PCCRC definition most commonly used was a CRC detected within 60 months of a colonoscopy. The quality of most studies was evaluated as high (Appendix A), and all were cohort studies. The characteristics of the included studies are shown in Table 2. The largest study that investigated the MMR status included 725 PCCRCs, and the largest study with genomic analysis included 82 PCCRCs. The molecular features analysed were the presence of MSI, CIMP, and CIN, and the most frequent mutations determined were *BRAF* and *KRAS*. In reviewing the literature, no data were found on the molecular characteristics of interval cancers according to CMS classification.

PCCRCs and clinically detected CRCs appear to have different biological characteristics, particularly regarding the presence of MSI. Among the six studies that analysed differences in the cancer incidence according to the MMR status, five reported that more PCCRCs displayed DNA mismatch repair deficiency than detected tumors. Some studies also showed that interval cancers were more frequent in the proximal colon. Additionally, MSI tumors were more likely than MSS cancers to occur in the proximal colon. These results, and the accelerated growth of small adenomas to invasive cancers due to MSI [86,87], could explain de novo rapid-growth neoplasias.

Two of the three studies that analysed CIMP found that interval cancers were more frequently CIMP-high. Furthermore, Arain et al. [77] and Sammadder et al. [84] stated that CIMP-high tumors were more likely to occur in the proximal colon and were MSI. CIN has only been analysed in the population of Winnipeg, and no association was found with PCCRC, a finding that is consistent with CIN being a prominent feature of distal cancers [88,89].

Studies evaluating the association between PCCRCs and the presence of mutations have been negative. Only Shaukat et al. [79] found an inverse association of *KRAS* mutation with interval cancer and the MSI status. Despite the known association in sporadic CRCs between *BRAF* mutation and CIMP-high, none of the studies have shown its association with PCCRCs. However, *BRAF*-mutated cancers were more likely to be MSI, poorly differentiated, mucinous in histology and located in the proximal colon [78]. Moreover, *BRAF* mutations and CIMP-high frequently coexisted [84].

In the English Bowel Cancer Screening Programme, Walsh et al. [49] investigated the characteristics of screen-detected and guaiac faecal occult blood test interval CRCs. They wanted to explore whether improved outcomes in screen-detected patients were explained by biological differences. Interval cancers were larger and more frequent with venous invasion. However, they found no differences in the proportion of MSI-high tumors, vascularity or the tumor growth rate. They concluded that the low sensitivity of the guaiac faecal occult blood test for CRC explained more interval CRCs than rapid growth and/or reduced bleeding.

## 4. Discussion

The interval CRC proportion identified in screening using the FIT in population-screening programmes is approximately 15%, slightly lower in successive screening rounds. According to clinical characteristics, interval cancers are more likely to arise in women and to be located in the right colon. At the molecular level, studies have shown that interval cancer is MSI and CIMP-high.

The reported molecular features of PCCRC implicate the role of at least two specific carcinogenic pathways, the MSI and serrated pathways. Additionally, overlapping of different carcinogenic pathways occurs. These differences reflect the molecular characteristics already described in right-sided tumors [89]. However, insufficient data exist to differentiate whether these characteristics are associated with missed tumors or de novo CRCs. Studies have also shown genetic similarities between clinically detected CRCs and PCCRCs, suggesting that interval CRCs could arise from missed sporadic lesions. The aetiology of interval PCCRC is thought to be explained mostly by missed or incompletely resected polyps on colonoscopy with some contribution from fast-growing new lesions [90,91,92]. Only one study performed in a stool-based screening programme using a guaiac-based test assessed the molecular characteristics of interval CRC. Compared with guaiac-based tests, FIT interval cancer might have fewer false-negative cancers and a higher frequency of fast-growing advanced neoplasias given its higher sensitivity for advanced adenomas and CRCs [3,93]. However, compared with PCCRC, more pedunculated lesions and distal polyps are expected among false-negative cancers. In summary, as the sensitivity of a screening test increases, the number of false-negatives decreases, and the proportion of fast-growing tumors increases.

Sessile serrated polyps are lesions more frequently missed, [94] with a higher rate of incomplete resection during colonoscopy [95] and a lower susceptibility to bleed. The reasons could be due to their proximal location, small size, flat or sessile morphology, and endoscopic appearance (mucus cap and irregular shape) [95]. Thus, sessile serrated polyps could be responsible for both PCCRCs and FIT interval cancers. Better understanding of the disease biology of serrated polyps is needed to improve surveillance strategies focusing on high-risk population. A strong relationship between serrated polyps and lifestyle factors (particularly smoking) has been reported in the literature [96]. Consistent with this finding, in a recent meta-analysis, smoking increased the risk of CRC characterized by MSI, CIMP-high, and BRAF mutation [97]. Taken together, these data show that smokers are susceptible to the development of interval cancers and may need a personalized strategy of post-polypectomy surveillance.

Regarding CMS categorization, although no study has been reported in interval CRC, a higher proportion of screen-detected cancers could be hypothesized to be CSM2 and CMS3 because the tumors have an adenoma-carcinoma 10-year interval and *KRAS* mutations. By contrast, interval cancers would be classified mostly in CSM1/CMS4. These two categories are good candidates to explain interval cancers for several reasons. First, both types are proposed to arise from serrated lesions [98]. However, CMS1 tumors mainly comprise MSI tumors enriched in the interval tumor setting [77]. Additionally, CMS4 tumors are aggressive and frequently detected in late stages. Thus, they could be hypothesized to have rapid growth that, in turn, might prevent them from being detected in a two-year screening round.

Despite the heterogeneity of the studies that molecularly characterized interval cancer and its small sample size, they all point in the same direction. However, further research is needed, particularly regarding FIT screening, because we cannot translate these data to interval CRC in FIT-based CRC screening. Although the study of Walsh et al. [49] was performed in a screening programme, it used guaiac faecal occult blood test as the screening test. Another limitation is that some of the studies in Table 2 did not exclude patients with hereditary CRC. This observation is critical given that the most important molecular feature analysed is the MMR status, which is characteristic of Lynch syndrome. However, the higher median age of subjects in the studies indicates that most MSI cancers were sporadic. Additional studies restricted to average-risk populations and in an FIT screening programme setting are needed.

An FIT interval CRC could be explained by FIT sensitivity and factors that alter it. However, in some cases, it could account for its own tumor characteristics (e.g., non-bleeding or intermittent-bleeding neoplasia or de novo rapid growth CRC). The ability to differentiate false-negative FIT results from true interval cancers would be very helpful. For example, if most of the interval CRCs are false-negative results, we should investigate the underlying factors (e.g., temperature [99,100] and subsequently implement improvements in screening programmes (e.g., maintaining screening cold chains). However, if interval cancers are rapid-growth neoplasias, efforts should be focused on developing new screening strategies, such as stratifying CRC risk using the previous haemoglobin concentration, age or/and sex. [101] Another example of better understanding the altered biology of interval cancers to optimize screening programmes and post-polypectomy surveillance recommendations would be to offer a simple recommendation to quit smoking. This finding is consistent with the already mentioned relationship between cigarette smoking and interval cancer molecular characteristics (MSI-high and serrated polyps).

## 5. Conclusions

Interval cancers are more likely to arise in women, to be located in the right colon, to display a DNA mismatch repair deficiency and to be CIMP-high. Knowing the characteristics associated with interval CRC as opposed to screen-detected CRC could improve our understanding of the carcinogenic pathways and could prevent interval CRC through risk stratification.

## Figures and Tables

**Figure 1 cancers-13-01328-f001:**
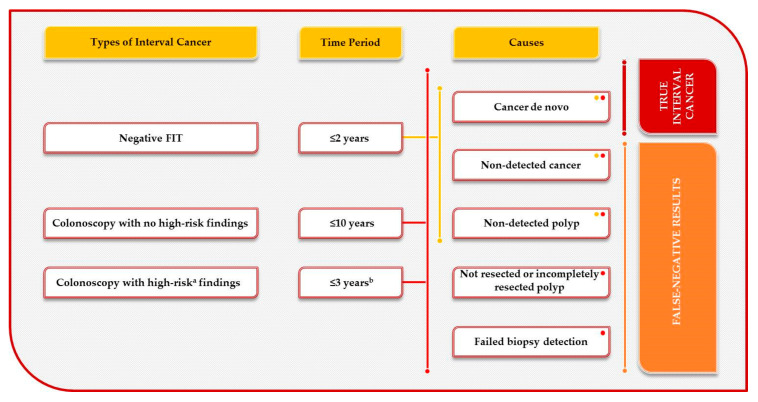
Causes of colorectal interval cancers. FIT interval cancer is caused by de novo cancer (true interval cancer) or non-detected lesions (false-negative FIT results). A post-colonoscopy colorectal cancer is also caused by de novo cancer (true interval cancer) or missed lesions, incompletely resected lesions and failed biopsy (false-negative results). ^a^ High-risk findings: individuals with complete removal of at least 1 adenoma ≥10 mm or with high-grade dysplasia, or ≥5 adenomas, or any serrated polyp ≥10 mm or with dysplasia according to current guidelines [5,6,7]. ^b^ Except for the need for a 2- to 6-month early repeat colonoscopy following piecemeal endoscopic resection of polyps ≥20 mm.

**Figure 2 cancers-13-01328-f002:**
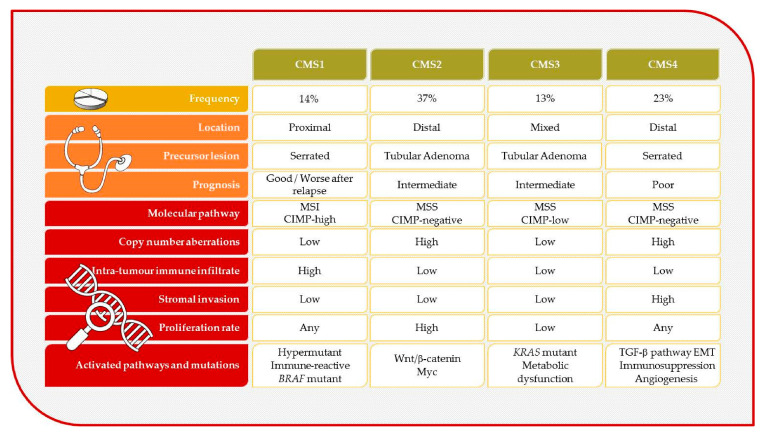
Characteristics of the molecular subtypes of colorectal cancer. CIMP: CpG island methylator phenotype; CMS: consensus molecular subtype; EMT: epithelial–mesenchymal transition; MSI: microsatellite instability; MSS: microsatellite stable.

**Table 1 cancers-13-01328-t001:** FIT sensitivity for CRC and epidemiological and clinical characteristics associated with interval CRC.

Author and Year	Country	Study Period	Number of Screening Sounds	FIT Cut-off, μg Hb/g Faeces	Number of Participants with a Negative FIT	Total Number of Screen-Detected CRCs	Total Number of iCRCs	iCRC, % (*n*)		FIT Sensitivity (%)		iCRC Characteristics
				Number of Samples				Men	Women	Interval Cancer Proportion (%) ^a^	Proportional Interval Cancer Rate ^b^ (95% CI ^c^)	
Parente et al., 2013 [37]	Italy	2005–2007	1 (Prevalent)	20	78,226	165	8	ND	ND	Total: 4.6	11.0 (5,22) ^d^	ND
				**1**	Aged 50–69 y							
Shin et al., 2013 [38]	Korea	2004–2007	2 (Prevalent and incident)	10	8,134,104	2961	2047	60.5 (1233)	39.4 (805)	Total: 59.1	25.0 (24,26) ^d^	Female
										1st round: 59.7		Proximal location
				1	Aged ≥ 50 y					Subsequent: 56.1		
Chiu et al., 2015 [39]	Taiwan	2004–2009	3 (Prevalent and incident)	20	1,113,932	2728	968	ND	ND	Total: 26.2	14.0 (14,15) ^d^	Female
				1	Aged 50–69 y							Advanced stages
Jensen et al., 2016 [40]	USA	2007–2012	4 (Prevalent and incident)	20	780,577	958	242	ND	ND	Total: 20.2	9.0 (8–11) ^d^	Advanced stages
				1	Aged 50–70 y							Proximal location
Giorgi Rossi et al., 2017 [41]	Italy	2000–2008	1 (Prevalent)	20	805,914 ^e^	ND	172	52.9 (91)	47.1 (81)	ND	21.0 (18,25) ^d^	Female
				1	Aged 50–69 y							Proximal location
Portillo et al., 2017 [42]	Spain	2009–2015	3 (Prevalent and incident)	17–20	769,124	2518	186	67.2 (125)	32.8 (61)	Total: 6.9	4.0 (3,5) ^d^	Proximal location
				1	Aged 50–69 y					1st round: 8.1		Advanced stages
										Subsequent: range 1–5		Worse prognosis
van der Vlugt et al., 2017 [43]	Netherlands	2006–2015	3 (Prevalent and incident)	10	15,711	116	27	59.0 (16)	41.0 (11)	Total: 22.3	24.0 (17,35) ^d^	Advanced stages
				1	Aged 50–74 y							
Burón et al., 2018 [44]	Spain	2010–2013	2 (Prevalent and incident)	20	161,691	415	92	17.6 (54)	18.8 (38)	Total: 18.0	ND	Advanced stages
				1	Aged 50–69 y					1st round: 16.0		
												Proximal location
										2nd round: 22.0		Rectum
Novak Mlakar et al., 2018 [45]	Slovenia	2011–2012	1 (Incident)	20	236,801	493	79	63.3 (50)	36.7 (29)	Total: 13.8	11.6	Proximal location
				2	Aged 50–69 y							Advanced stages
van der Veerdonk et al., 2019 [46]	Belgium	2013–2017	2 (Prevalent and incident)	15	1,123,479	4094	772	54.0 (417)	46.0 (355)	Total: 18.9	ND	Female
				1	Aged 56–74 y					1st round: ND ^f^		Older age
										2nd round: 5.4		Proximal location
Toes-Zoutendijk et al., 2020 [47]	Netherlands	2014–2018	1 (Prevalent)	15 and 47	111,800 and 373,174 ^g^	1102 and 2108 ^g^	126 and 418 ^g^	42.1 and 50.0 ^g^ (73 and 209 ^g^)	57.9 and 50.0 ^g^ (53 and 209 ^g^)	Total: 10.3 and 16.5 ^g^	9.5 and 13.8 ^g^	Female
				1	Aged 55–75 y							Older age
												Proximal location
Zorzi et al., 2020 [48]	Italy	2002–2015	6 (Prevalent and incident)	20	423,539	781	150	48.0 (72)	52.0 (78)	Total: 16.1	13.7 (12–16)	Female
										1st round: 11	1st round: 18.5 (14–24)	
				1	Aged 50–69 y					Subsequent range: 7–27	Subsequent range: 8–16	Proximal location

CRC, colorectal cancer; FIT, faecal immunochemical test; Hb, haemoglobin; iCRC, interval colorectal cancer; NA: not applicable; ND, not described; y: years. ^a^ Interval cancer proportion = [Interval cancers/(interval cancers + cancers screen-detected)] × 100; ^b^ Proportional interval cancer rate = (Observed interval cancers/Expected incident cancers) × 100; ^c^ If data were available.; ^d^ Data reported in Wieten et al. [35]; ^e^ 579,176 and 226,738 persons with negative test results followed up at 12 and 24 months, respectively.; ^f^ Data reported in van der Veerdonk et al., 2019 [46] with errata.; ^g^ FIT cut-offs of 15 μg Hb/g faeces and 47 μg Hb/g faeces, respectively.

**Table 2 cancers-13-01328-t002:** Studies that molecularly characterized interval cancer.

Author and Year	Setting (Country)	Definition of iCRC	N	Exclusion for	Matching	Interval Cancer Characteristics Compared with Detected Cancers				Other iCRC Characteristics
						MSI	CIMP	CNI	Mutations	
Sawhney et al., 2006 [76]	Minneapolis Veterans Administration Population (USA)	PCCRC: CRC within 60 months of a colonoscopy	51 PCCRCs, 112 detected	IBD, FAP	Matched 2:1 by sex and age	↑ 30% PCCRCs vs. 10% DCRCs; OR^a^: 3.7; 95% CI, 1.5–9.1	NA	NA	NA	PCCRCs were more proximal, smaller and had mucinous histology. No differences for histologic grade and TNM stage.
Arain et al., 2010 [77]			52 PCCRCs, 103 detected (including patients from 2006 study)			↑ 29% PCCRCs vs. 11% DCRCs; OR ^b^: 2.7; 95% CI, 1.1–6.8	↑ 57% PCCRCs vs. 33% DCRCs; OR ^b^: 2.4; 95% CI, 1.2–4.9	NA	NA	
Shaukat et al., 2010 [78]		DCRC: CRC on index colonoscopy	63 PCCRCs, 131 detected (including patients from 2006 and 2010 study)			↑ 29% PCCRCs vs. 11% DCRCs; OR ^b^: 2.6; 95% CI 1.1–6.7	NA	NA	*KRAS* ↓ 13% PCCRCs vs. 29% DCRCs; OR ^b^: 0.4; 95% CI, 0.2–0.9; *BRAF* = 28% PCCRCs vs. 19% DCRCs; OR ^b^: 0.9; 95% CI, 0.4–2.4	
Shaukat et al., 2012 [79]										
Nishihara et al., 2013 [75]	Nurses’ Health Study/Health Professionals Follow-up Study (USA)	PCCRC: CRC within 60 months of a colonoscopy	62 PCCRCs, 606 detected	IBD, FAP	No	↑ 25% PCCRCs vs. 14% DCRCs; OR ^c^: 2.1; 95% CI, 1.1–4.0	↑ 30% PCCRCs vs. 15% DCRCs; OR ^c^: 2.2; 95% CI, 1.1–4.2	NA	*BRAF* = 22% PCCRCs vs. 14% DCRCs; OR ^b^: 1.8; 95% CI, 0.3–3.6; *KRAS* = 23% PCCRCs vs. 36% DCRCs; OR ^b^: 0.6; 95% CI, 0.3–1.1	High-level long interspersed nucleotide element-1 (LINE-1) methylation (OR ^c^: 3.21; 95% CI, 1.3–8.0). *PIK3CA* mutation was not associated.
	**Opportunistic screening with colonoscopy**	DCRC: CRC >60 months of a colonoscopy or without prior endoscopy								
Richter et al., 2014 [80]	Massachusetts General Hospital (USA)	PCCRC: CRC within 12–60 months of a colonoscopy	42 PCCRCs (42 by MMR status and genomic analysis)	IBD; FAP	No	41% of PCCRCs	NA	NA	*BRAF* = 17% PCCRCs vs. 10% DCRCs ^d^; *KRAS* = 29% PCCRCs vs. 37% DCRCs ^d^	*NRAS**,**PIK3CA,**APC, CTNNB1, EGF, PTEN,* and *TP53* mutations were not associated.
		DCRC: CRC on index colonoscopy	226 detected (226 with only genomic analysis)							
Cisyk et al., 2015 [81]	Winnipeg Population-based (Canada)	PCCRC: CRC within 6–36 months of a colonoscopy	46 PCCRC (46 analysed)	IBD < 50 years	Matched 2:1 by sex, age and tumor location	NA		= 82% PCCRCs vs. 85% DCRCs ^d^	NA	No differences in location
			95 detected (95 analysed)							
Cisyk et al., 2018 [82]		DCRC: CRC within 1 month of a colonoscopy	46 PCCRCs (45 analysed)			↑ MSI~1.5x higher ^d^ 27% PCCRCs vs. 17% DCRCs	NA	= 14% PCCRCs vs. 12% DCRCs CIN^+^/MSI^+d^	NA	
			95 detected (90 analysed)							
Lee et al., 2016 [83]	Kangbuk Samsung Hospital (Korea)	PCCRC: CRC within 12–60 months of a colonoscopy	Only CRC removed surgically	IBD; Hereditary cancer	No	↑ 32% PCCRCs vs. 8% DCRCs; OR ^e^: 3.9; 95% CI: 1.3–11.0	NA	NA	NA	No differences in age, sex, location or TNM.
		DCRC: CRC index colonoscopy or >60 months of a colonoscopy	25 PCCRCs, 261 detected							
Stoffel et al., 2016 [15]	Denmark Population-based (69% of CRC nationwide) (Denmark)	PCCRC: CRC within ≥ 6 months of a colonoscopy	725 PCCRCs (725 by MMR status and 85 by genomic analysis)	None	No	30% PCCRCs vs. 17% DCRCs; OR: 1.4; 95% CI: 0.9–2.1	NA	NA	19% *BRAF* mutations in PCCRCs; 27% *KRAS/ NRAS* mutations in PCCRC	PCCRCs were associated with older patients, earlier stage, IBD and proximal tumors. 19% *P1K3CA* mutations in PCCRCs.
		DCRC: CRC within 6 months of a colonoscopy	9640 detected (8337 by MMR status and no genomic analysis)							
Walsh et al., 2016 [49]	The Northern Region Colorectal Cancer Audit Group cohort (United Kingdom) **Stool-based CRC screening programme**	iCRC: after a negative gFOBT and before next invitation	28 iCRCs	IBD; Hereditary cancer	Matched 1:2–3 by tumor location, Dukes’ stage, and histological differentiation grade	= 14% iCRCs vs. 5% DCRCs ^d^	NA	NA	NA	iCRCs were larger and more likely to exhibit histological venous invasion.
		Screen-DCRC: after a positive gFOBT	43 screen-DCRCs							
Samadder et al., 2019 [84]	Utah Population-based (USA)	PCCRC: CRC within 6–60 months of a colonoscopy	Clinical data: 159 PCCRCs (84 analysed);	None	Matched by age, sex, andhospital site	↑ 32% PCCRCs vs. 13% DCRC; OR ^f^: 4.2; 95% CI: 1.5–11.1	= 17% PCCRCs vs. 21% DCRC; OR ^f^: 1.4; 95% CI: 0.6–3.2	NA	*BRAF* = 24% PCCRCs vs. 24% DCRCs; OR ^f^: 1.2; 95% CI: 0.6–2.4; *KRAS* = 30% PCCRCs vs. 29% DCRCs; OR ^f^: 1.1; 95% CI: 0.5–2.2	PCCRCs were associated with proximal colon and early stage.
		DCRC: CRC within 6 months of a colonoscopy	2500 DCRC (84 analysed)							
Tanaka et al., 2020 [85]	Single-centre Hiroshima University Hospital (Japan)	PCCRC: CRC or pTis within 6–60 months of colonoscopy	34 PCCRCs (33 by MMR status and 23 by genomic analysis. Note: 18 of 33 were pTis)	Hereditary Cancer; IBD	No	21% of PCCRCs	NA	NA	13% *BRAF* mutations in PCCRCs; 52% *KRAS/NRAS* mutations in PCCRCs	22% *P1K3CA* mutations in PCCRCs.
		DCRC: CRC within 6 months of a colonoscopy.	1698 DCRCs (not analysed)							

NOTE. Bold indicates studies that were performed in screening populations. The grey shadow refers to the direction of the association between the factor and interval cancers, only shown when the study compares PCCRCs with DCRC cancers. ^a^ Adjusted by age; ^b^ Adjusted by size, location, and histology differentiation; ^c^ Adjusted by body mass index, smoking status, familiar history of CRC, aspirin, physical activity level, red meat intake, caloric intake, alcohol intake, folate intake, calcium intake, multivitamin use, nonsteroidal antiinflammatory drug use, and cholesterol-lowering drug use; ^d^ OR and/or 95% not available; ^e^ Adjusted by age, MSI, and location; ^f^ Adjusted by CIMP, MSI, *BRAF,* and *KRAS*. CIMP: CpG island methylator phenotype; CMS: consensus molecular subtype; CNI: chromosomal number instability; CRC: colorectal cancer; DCRC: detected colorectal cancer; FAP: Familial adenomatous polyposis; IBD: inflammatory bowel disease; iCRC: interval CRC; gFOBT: guaiac faecal occult blood test; MMR mismatch repair; MSI: microsatellite instability; NA: not available; PCCRC: post-colonoscopy colorectal cancer.

## Data Availability

Not applicable.

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
