# Peer review of "Future Prospects of Colorectal Cancer Screening: Characterizing Interval Cancers"

_cancers, 2021, doi:10.3390/cancers13061328_

Round 1
Reviewer 1 Report
This is a much improved version of the manuscript
Reviewer 2 Report
The article seems to be a narrative review, albeit the authors did not state clear objectives and did not describe the methodology they followed. In my opinion, lack of compliance with current recommendations for the preparation of a narrative review in clinical research makes the outcomes of this review not measurable and the information may be less accessible to the larger public.
Author Response
While we appreciate the reviewer’s feedback, we respectfully disagree. We have followed the 6 essential points of the Scale for the Assessment of Narrative Review Articles (SANRA) [Baethge et al. Res Integr Peer Rev. 2019 Mar 26;4:5. doi: 10.1186/s41073-019-0064-8. eCollection 2019] to prepare the review. The following comments clarify where each SANRA question is explained:
- Justification of the article’s importance for the readership: Lines 80-97
- Statement of specific aims or formulation questions: Lines 195-201
- Description of the literature search: Lines 205-241
- Referencing: See all the manuscript where every statement has been backed by references.
- Scientific reasoning: We have mentioned the study design of all the studies included in the review. We have assessed the quality of the studies included in the review with the Newcastle-Ottawa Scale [Wells et al. The Newcastle-Ottawa Scale (NOS) for assessing the quality of nonrandomised studies in meta-analyses. 2010].
- Appropriate presentation of data: In order to summarize the important data, we have constructed Table 1 and Table 2. Effect sizes are accompanied by confidence intervals.
As Reviewer#3 commented, we think this study makes a valuable contribution to the field because it summarises the accumulated state of knowledge on interval CRCs. It critically evaluates material that has already been published. Identification of clinical and molecular characteristics associated with interval CRC, in contrast to screen-detected CRC, may provide new insights into how to prevent interval CRC more effectively through improved risk stratification.
Reviewer 3 Report
The Review article Authored by Dr Ibáñez-Sanz et al aims to summarize the current knowledge of the epidemiological, clinical and molecular characteristics of colorectal interval cancers. Colorectal interval cancers are defined as tumors that are diagnosed clinically after a negative result of screening tests (biennial faecal immunochemical test FIT, or within 10 years post-colonoscopy). Their origin might be attributed to a missed lesion, a not resected or incompletely resected lesion (colonoscopy), or a de novo rapid growth CRC.
After Introducing these elements, the manuscript provides an extensive analysis of the literature to delineate 1) the proportion and epidemiological and clinical characteristics of interval CRC, and 2) their molecular characteristics. This later chapter concerns essentially post-colonoscopy CRC, since only one study reports the molecular characterization of interval cancers after FIT.
This Review article revealed that the proportion of interval cancer using FIT screening program is 15%, but decreased after successive round of screening. Furthermore, interval CRC arises more frequently in women, are preferentially located in right colon and characterized by MSI and CIMP. This study also highlights the requirement to investigate further the molecular characteristics of interval CRC in the context of FIT program screening. Accordingly, a FIT interval CRC might result from FIT sensitivity, but also to the intrinsic characteristics of the tumor (e.g., non-bleeding, location or de novo rapid growth CRC). Thus, the ability to differentiate false-negative FIT from true interval cancers would contribute to improve screening strategies. This manuscript is well written and constitutes an up-to-date analysis of the literature concerning colorectal interval cancers. This topic is of importance considering that early detection of colorectal lesions leads to improving patient outcomes while reducing the overall cost of care. This review would make therefore a suitable contribution for “Cancers”.
Minor points
Table 2 The portrait format of the 1st page induces a truncation of the last 3 columns
Few typos need to be fixed, e.g line 103 “conformation” vs “confirmation”
Author Response
We would like to express our great appreciation to you. Thank you very much for your previous comments that helped us improve this manuscript.
Minor points
- Table 2 The portrait format of the 1st page induces a truncation of the last 3 columns
Thank you for pointing this out. We have reduced the number of words by eliminating expendable connectors and including two abbreviations: Familial adenomatous polyposis (FAP) and detected colorectal cancer (DCRC). We have erased the sentence “No differences in ulcerating morphology, microvessel density, and proliferation index” of the “Other iCRC characteristics” column of the Walsh et al. study because it was already stated in line 422.
We have revised Table 2 in order to avoid the presence of truncated words.
Few typos need to be fixed, e.g line 103 “conformation” vs “confirmation”
Thanks for noticing it. We have corrected it and we have carefully revised the manuscript in order to identify any more typos. Another typos and mistakes have been corrected and marked using Tracked changes.
This manuscript is a resubmission of an earlier submission. The following is a list of the peer review reports and author responses from that submission.
Round 1
Reviewer 1 Report
I have read with great interest the article entitled “Future prospects of colorectal cancer screening: molecular characterization of interval and screen-detected tumours”. However, it has some relevant limitations that authors should solve.
- The structure of the article is difficult to follow. It does not have a introduction that structures all the review and the aims of the review.
- First point: The authors should clarify the definition of interval cancer in the sense that it also includes a CRC detected before the scheduled colonoscopy surveillance. For a reader not used to the definition, the second paragraph is not clear.
- In the fourth paragraph, they should add some relevant information: Meta-analysis on the diagnostic accuracy of fecal immunochemical tests in CRC screening (32620599) and some relevant articles regarding the characteristics of adenomas not detected by FIT (24962836)
- The authors include a systematic review on interval CRC rate but they do not provide any information regarding quality and bias of the articles evaluated. It is a shame as long as there is not much information regarding this point.
- With respect to the explanation provided by U.Haug in her article to the lower FIT sensitivity for advanced proximal neoplasia and advanced neoplasia in female and older patients is difficult to sustain. It is much more pausible that proximal CRC arise from advanced serrated lesions or advanced flat/sessile proximal adenomas that are not detected by FIT. These lesions are related to the serrated pathways that explains proximal CRC in old population, mainly females, as the authors comment in the next chapter. To contrast, there is evidence that FIT sensitivity for CRC is not modified by location in symptomatic patients (Cubiella et al Colorectal Disease 2014).
- Although the chapter regarding molecular characterization of tumors Is interesting, it should be clearly improved:
- The description of the consensus molecular subtypes should be reduced, it is too extensive.
- The authors do not include relevant information regarding molecular characteristics of interval CRC (Cisyk et al Dig Dis Sci 2014: Sawhney, Gastroenterology 2006)
- The definition used in the described molecular studies for interval CRC was what we now call postcolonoscopy CRC. So, we can not translate this information to interval CRC in FIT based CRC screening. These cancers are a mixture of FIT false negative, fast transformation of preneoplastic lesions and new onset CRC.
- The first paragraph of the third page is not related to interval CRC but rather to hypothesize what is the most probable CMS categorization on interval CRC (no evidence at all) and the use of genomic expression analysis in CRC. I encourage the authors to delete this paragraph as long as it adds no information to the review.
- Accordingly to my previous comments, authors should modify the conclusions.
- The article requires English editing.
Author Response
Reviewer #1:
I have read with great interest the article entitled “Future prospects of colorectal cancer screening: molecular characterization of interval and screen-detected tumours”. However, it has some relevant limitations that authors should solve.
- The structure of the article is difficult to follow. It does not have an introduction that structures all the review and the aims of the review.
The article was initially intended as a minireview/commentary article. Following this recommendation, we have restructured the manuscript as a review. It summarizes the evidence of the epidemiological, clinical and molecular characteristics of CRC interval cancers. For that purpose, first, we reviewed the interval CRC proportion identified in FIT population-based screening programmes and summarized patient-related characteristics associated with interval cancer incidence rate. Secondly, we have summarized all the evidence concerning the molecular characterization of interval CRC.
- First point: The authors should clarify the definition of interval cancer in the sense that it also includes a CRC detected before the scheduled colonoscopy surveillance. For a reader not used to the definition, the second paragraph is not clear.
We agree with the reviewer. We have rewritten the paragraph and changed Figure 1 to clarify this point.
- In the fourth paragraph, they should add some relevant information: Meta-analysis on the diagnostic accuracy of fecal immunochemical tests in CRC screening (32620599) and some relevant articles regarding the characteristics of adenomas not detected by FIT (24962836)
We thank the reviewer for this suggestion; we have added a paragraph with all this value information.
- The authors include a systematic review on interval CRC rate but they do not provide any information regarding quality and bias of the articles evaluated. It is a shame as long as there is not much information regarding this point.
The study quality score of the observational studies has been analysed according to the Newcastle-Ottawa (Wells G et al, 2010) criteria for each review. We have added two supplementary tables with this information.
- With respect to the explanation provided by U.Haug in her article to the lower FIT sensitivity for advanced proximal neoplasia and advanced neoplasia in female and older patients is difficult to sustain. It is much more pausible that proximal CRC arise from advanced serrated lesions or advanced flat/sessile proximal adenomas that are not detected by FIT. These lesions are related to the serrated pathways that explains proximal CRC in old population, mainly females, as the authors comment in the next chapter. To contrast, there is evidence that FIT sensitivity for CRC is not modified by location in symptomatic patients (Cubiella et al Colorectal Disease 2014).
Following Reviewer#1 recommendation, we have rewritten the Results section of the manuscript regarding the above information. We have added new references.
Although the chapter regarding molecular characterization of tumors is interesting, it should be clearly improved:
- The description of the consensus molecular subtypes should be reduced, it is too extensive.
Done!
- The authors do not include relevant information regarding molecular characteristics of interval CRC (Cisyk et al Dig Dis Sci 2014: Sawhney, Gastroenterology 2006)
We have now included a Table that summarizes and compares all the studies that have molecularly characterised interval cancers. We believe this table greatly improves this second part of the review.
- The definition used in the described molecular studies for interval CRC was what we now call postcolonoscopy CRC. So, we cannot translate this information to interval CRC in FIT based CRC screening. These cancers are a mixture of FIT false negative, fast transformation of preneoplastic lesions and new onset CRC.
In order to clarify this, a better description of each study has been added in Table 2. Moreover, we have further discussed this issue in the Discussion.
- The first paragraph of the third page is not related to interval CRC but rather to hypothesize what is the most probable CMS categorization on interval CRC (no evidence at all) and the use of genomic expression analysis in CRC. I encourage the authors to delete this paragraph as long as it adds no information to the review.
We partly agree with the reviewer. We have moved this paragraph to the Discussion and we have contextualized the information.
- Accordingly to my previous comments, authors should modify the conclusions.
We absolutely agree with this comment. We have modified the title, the abstract and the conclusions accordingly.
- The article requires English editing.
Following this recommendation, the manuscript has been carefully reviewed by an English native speaker.
Reviewer 2 Report
This manuscript aims to discuss the molecular characteristics of interval and screen detected colorectal cancers within the context of screening.
Whilst this is an important subject that deserves to be highlighted to facilitate enhanced early detection of CRC, the current content and perspectives are insufficient for an impactful review, and would be more appropriate as a short correspondence. The text comes across as a list of previously published facts with insufficient discussion, interpretation and reflection. The manuscript is difficult to read in places which is distracting for the reader. The figures do little to illustrate the points made in the text.
Author Response
Reviewer #2:
This manuscript aims to discuss the molecular characteristics of interval and screen detected colorectal cancers within the context of screening. Whilst this is an important subject that deserves to be highlighted to facilitate enhanced early detection of CRC, the current content and perspectives are insufficient for an impactful review, and would be more appropriate as a short correspondence. The text comes across as a list of previously published facts with insufficient discussion, interpretation and reflection. The manuscript is difficult to read in places which is distracting for the reader. The figures do little to illustrate the points made in the text.
As suggested, the manuscript has been extensively revised in a way we hope is clearer for the reader. We believe that now is an overview of all the epidemiologic, clinical and molecular characteristics of interval CRC. We have restructured it in order to improve its readability. Figure 1 has been modified so that it includes the new recommendations of post-polypectomy surveillance. Figure 2 has been improved and used to eliminate some information about CMS. We think this figure could be useful for readers unfamiliar with CMS. We have included a Table that summarizes all the evidence of molecular characteristics. Finally, we have further discussed and interpreted the review findings.